# Work Engagement and Burnout in a Private Healthcare Unit in Greece

**DOI:** 10.3390/ijerph21020130

**Published:** 2024-01-25

**Authors:** George Alexias, Maria Papandreopoulou, Constantinos Togas

**Affiliations:** 1Department of Psychology, Panteion University of Social and Political Sciences, 17671 Athens, Greece; galexias@panteion.gr; 2School of Social Sciences, Postgraduate Programme “Health Care Management”, Hellenic Open University, 26335 Patras, Greece; std130776@ac.eap.gr

**Keywords:** work engagement, burnout, health professionals, private sector, healthcare unit

## Abstract

Work engagement represents a positive work-related state of mind characterized by three dimensions: Vigor (high levels of energy and mental resilience during work), Dedication (strong involvement in one’s work), and Absorption (complete-joyous immersion in one’s tasks). This study aimed to investigate work engagement and burnout in health professionals in a private healthcare unit in Greece. A cross-sectional study was conducted with a sample of 151 professionals, including doctors, nurses, administrative staff, and other health professionals involved in this setting. The study duration was four months (January–April 2022). The questionnaire covered sociodemographic and work-related information, along with the Utrecht Work Engagement Scale and the Maslach Burnout Inventory. Regarding the subdimensions of the Utrecht Work Engagement Scale, participants demonstrated a moderate score in Absorption and medium to high scores in Vigor and Dedication. In terms of burnout, they showed a low score in Depersonalization, a medium score in Emotional Exhaustion, and a high score in Personal Accomplishment. Those with nonpermanent employment contracts scored higher in all dimensions of burnout compared to those with permanent employment contracts. Vigor, Dedication, and Absorption correlated negatively with Emotional Exhaustion and Depersonalization and positively with Personal Accomplishment. Vigor negatively predicted Emotional Exhaustion, Depersonalization, and Personal Accomplishment. In conclusion, healthcare professionals in the private healthcare sector in Greece demonstrate moderate work engagement and experience moderate burnout. There are differences in work engagement and burnout based on sociodemographic and work-related characteristics. Promoting work engagement (especially the dimension of Vigor) is essential to preventing and addressing burnout among healthcare professionals.

## 1. Introduction

Healthcare workers face a heightened risk of occupational burnout, due to factors such as demanding workloads, a perceived lack of control, and insufficient extrinsic and intrinsic job rewards. Conversely, the contribution of work engagement to the psychological well-being of workers is substantial [1,2]. Work engagement is defined as a fulfilling, work-related positive state of mind composed of three dimensions: Vigor, Dedication, and Absorption. Vigor denotes high levels of energy and mental resilience during work. Dedication is characterized by a strong involvement in one’s work and deriving a sense of significance from it, whereas Absorption refers to complete and joyous immersion in one’s tasks [3]. 

There is a disparity in reported levels of work engagement among healthcare professionals, which can be attributed to institutional variations, cultural contexts, and the diversity of healthcare roles. For instance, Engelbrecht et al. argue that healthcare professionals generally exhibit moderate to high levels of work engagement, with a pronounced emphasis on Dedication [4]. In contrast, a study conducted by the World Health Organization revealed that only 13% of workers worldwide are fully engaged in their work and that most nurses express dissatisfaction with their jobs. Furthermore, approximately 40% of health professionals, including doctors, nurses, and midwives, would contemplate leaving their current employment if they were dissatisfied with their work [5]. These findings align with those of other researchers who emphasize suboptimal engagement levels, especially in professions requiring substantial dedication and energy [6].

Women, doctors, nurses, and older individuals tend to report higher dedication compared to men, younger individuals, and other health professionals working in this setting. [7,8,9]. Concerning private health units, studies have indicated moderate to high levels of work engagement, with Dedication receiving the highest score followed by Absorption and Vigor [10].

Moreover, healthcare professionals on the front lines of the COVID-19 crisis were at high risk of psychological distress. In this case, work engagement could be a protective factor [11]. It is noteworthy that the COVID-19 pandemic imposed a substantial burden on the healthcare system of Greece, akin to other countries. According to official data, there were 5,363,920 cases and 37,370 deaths recorded in Greece from the onset of the pandemic until August 2023 [12].

Burnout has been identified as a syndrome comprising three dimensions: emotional exhaustion, depersonalization, and personal accomplishment. It is linked to adverse physical and mental health effects, resulting in low productivity, frequent absences from work, and diminished work engagement [13]. The role of health systems is crucial in addressing this syndrome. Although burnout may manifest in individuals, its fundamental roots lie within systems [14]. The primary factors contributing to employee burnout often have less to do with the individual and more to do with how they are managed. Therefore, there is a need to focus more on the social and organizational environments in which individuals work [15]. A related survey identified the top five reasons for burnout: unfair treatment at work, lack of role clarity, an unmanageable workload, insufficient communication and support from managers, and unreasonable time pressures [16].

Numerous studies have investigated burnout in healthcare professionals, revealing its impact on 10–70% of nurses, 30–50% of physicians, and at least 20% of Emergency Department and Intensive Care Unit workers [17,18,19,20,21,22]. 

A related study in Greece identified moderate levels of burnout and high emotional exhaustion. Work in specific departments, such as Pediatric and Psychiatric Clinic and the Emergency Department, was associated with higher emotional exhaustion. Additionally, individuals with many years of experience had lower levels of burnout, while men experienced higher depersonalization [23]. In contrast, in Katsiou’s research involving nurses, no differences were observed based on demographic or work characteristics [24]. Several studies conducted during the COVID-19 pandemic, involving samples of nurses, revealed moderate to high levels of burnout [25,26]. 

Various studies have explored the burnout of healthcare professionals in relation to their demographic characteristics. In a study conducted in Greece with a sample of nurses, men reported slightly lower levels of burnout, particularly in the dimension of personal accomplishment, whereas university graduates exhibited higher levels compared to secondary school graduates [26]. 

Several studies conducted in private healthcare units have revealed moderate to high levels of burnout, which were higher when compared to public health facilities. These studies involved samples of nurses, physicians, respiratory therapists, and other healthcare professionals [27,28,29]. Additionally, Al-Omari et al. found higher emotional exhaustion among women and nurses compared to individuals working in other professions [28]. 

Work engagement is positively associated with resilience and is considered the opposite of burnout [2,30]. Certainly, other researchers, such as Schaufeli and Bakker argue that work engagement and burnout are distinct concepts [31]. Work engagement plays a crucial role in addressing burnout, especially given the challenges that modern healthcare systems are currently facing [32]. According to Zeng and Chen, employees who experience burnout are more likely to have lower work engagement [32]. Conversely, burnout is reduced when employees have high levels of work engagement [33]. 

Work engagement and burnout among professionals appear to differ significantly between privately and publicly administered hospitals in various domains, including workload, job security, bureaucracy, goal setting, and opportunities for professional development [27,34]. 

In Greece, the majority of large hospitals fall under the public sector. Private hospitals are primarily concentrated in the capital city, Athens, and encompass both general and specialized facilities, such as maternity or psychiatric care clinics. Additionally, there are smaller private healthcare units, known as “clinics”, offering specialized therapeutic care such as general surgery or childbirth services. Similar to other countries, Greece displays significant differences in the organization, administration, and operation of public and private hospitals. Public hospitals provide primary, secondary, and tertiary healthcare, employing civil servants, whereas private hospitals mainly deliver secondary healthcare, and their professionals hold nonpermanent contracts or serve as visiting doctors. There are 24,412 doctors, 49,232 nurses, and 8268 other health professionals working in such settings with a nonpermanent employment status [35]. 

However, the relationship between work engagement and burnout has not been sufficiently studied in private healthcare professionals in Greece. This study aims to evaluate the work engagement and burnout of professionals in a private healthcare unit and to identify the demographic and work-related characteristics that affect them. These concepts can impact their work efficiency and the quality of care they provide [36]. Based on the literature review, the following research questions were formulated:Are there differences in the healthcare professionals’ work engagement, related to their sociodemographic characteristics? (research question 1) [7,8].Are there differences in their work engagement, based on their work-related characteristics? (research question 2) [7,8,9].Are there differences in the levels of their burnout, based on their demographic characteristics? (research question 3) [24,26,28].Are there differences in the levels of their burnout, based on their work-related characteristics? (research question 4) [23,24,27,28,29].Is there a correlation between work engagement and healthcare professionals’ burnout? (research question 5) [2,31,33,34,35].

## 2. Methods

### 2.1. Design Procedure

This study utilized a quantitative cross-sectional design with health professionals working in contact with patients: doctors, nurses, administrative staff, and other health professionals working in the setting. The setting was a private healthcare unit in the southern suburbs of Athens, Greece. This is a general healthcare unit consisting of 10 clinics. Participants received the questionnaires from the researcher and completed them either in their office or at home. The study duration was four months (January–April 2022). 

### 2.2. Participants

The sample comprised 151 employees (n = 151) working in the private healthcare unit mentioned above. The participants’ detailed demographic and work-related characteristics are presented in Table 1.

### 2.3. Measures

A questionnaire was utilized, comprising demographic and work-related information, the Utrecht Work Engagement Scale, and the Maslach Burnout Inventory. These scales have been translated and validated in the Greek population. 

#### 2.3.1. Demographic and Work-Related Information

In the first section of the questionnaire, participants responded to questions regarding their sex, age, marital status, level of education, number of children, and monthly income in Euros (categorized as up to 800 €, 801–1300 €, or >1301 €). They also provided information about their working status, including their profession (e.g., doctor, nurse), the type of employment contract (permanent or nonpermanent), and the number of years they had been working in the healthcare unit. 

#### 2.3.2. Utrecht Work Engagement Scale (UWES)

The scale evaluates the levels of mental resilience and energy during work, along with evaluating a sense of inspiration, challenge, pride, significance, and concentration in work [3]. It comprises 17 items divided into three subscales: (1) Vigor (6 items), e.g., “At my work, I feel bursting with energy”, (2) Dedication (5 items), e.g., “I am enthusiastic about my job”, (3) Absorption (6 items), e.g., “When I am working, I forget everything else around me”. Responses are rated on a 5-point Likert scale (1 = never, 2 = sometimes, 3 = often, 4 = very often, and 5 = always), with higher scores indicating higher work engagement. The mean score in each subscale is used for scoring. A short form of the Utrecht Work Engagement Scale has been translated into various languages and has demonstrated robust psychometric properties [37]. In this study, its Greek version was employed, and internal consistency reliability was very good to excellent (Cronbach’s Alpha value: Vigor = 0.914, Dedication = 0.907, and Absorption = 0.892) [38]. 

#### 2.3.3. Maslach Burnout Inventory (MBI)

It assesses occupational burnout and consists of twenty-two items, divided into the dimensions of emotional exhaustion, depersonalization, and personal accomplishment [30]. Emotional Exhaustion evaluates feelings of being emotionally exhausted by one’s work, e.g., “I feel frustrated by my work”. Depersonalization measures an impersonal and unfeeling response toward recipients of one’s service, treatment, or instruction, e.g., “I feel I look after certain patients/clients impersonally as if they are objects”. Personal Accomplishment measures feelings of competence and achievement in one’s work, e.g., “Through my work, I feel that I have a positive influence on people”. Responses are rated on a 7-point Likert scale (0 = never happens to me, 1 = a few times a year or less, 2 = once a month or less, 3 = a few times a month, 4 = once a week, 5 = a few times a week, and 6 = it happens to me every day). The score on each subscale results from the sum of the corresponding items. High scores in the Emotional Exhaustion or Depersonalization scales or low scores in the Personal Accomplishment scale indicate greater experienced burnout. This inventory is the most commonly used tool to measure burnout and it has been used in a variety of studies on health professionals (e.g., nurses, physicians, health aides, and health counselors, etc.). In this study, its Greek version was used [36]. Its internal consistency reliability was found to be good to excellent for all its subscales (Cronbach’s Alpha value: Emotional Exhaustion = 0.901, Depersonalization = 0.889, and Personal Accomplishment = 0.786). 

### 2.4. Data Analysis

The data analysis was conducted using IBM SPSS, version 26 (IBM, Armonk, NY, USA). Cronbach’s Alpha coefficient was employed to evaluate the internal consistency reliability of the scales. Values of Cronbach’s Alpha equal to or greater than 0.70 were considered indicative of good reliability. 

The normality of quantitative variables was examined using either the Kolmogorov–Smirnov test or the Shapiro–Wilk test. Descriptive statistics, including frequencies and percentages (%), were employed to summarize the data. Pearson’s correlation coefficient was used to explore linear correlations among quantitative variables.

To assess statistically significant differences between groups of categorical variables, *t*-test and Analysis of Variance (ANOVA) were utilized. A *t*-test was employed to compare the means of two groups and determine whether the observed differences were more likely to occur by random chance. ANOVA included various statistical models and associated estimation procedures to analyze differences among means. Post-hoc comparisons in ANOVA were conducted using the Bonferroni test. 

Additionally, three multiple regression analyses were performed to investigate how the work engagement dimensions and various work-related characteristics predict Depersonalization, Emotional Exhaustion, and Personal Accomplishment. The statistical significance level (*p*-value) was set at 5%.

### 2.5. Ethics

This study obtained approval from the Institutional Review Board of the Hellenic Open University. Each participant was given the option to participate after receiving information about the study’s purpose. Those who agreed to take part provided informed consent by signing the consent form before completing the questionnaire. Participants were assured of the anonymity and confidentiality of the information they provided. They were also informed that they could discontinue completing the questionnaire at any time if they wished to do so. The data collected were handled with the utmost protection and used exclusively for this study. 

## 3. Results

A total of 196 questionnaires were distributed, and 151 of them were returned. Therefore, the response rate for this study was 77%. 

At first, we performed descriptive analysis for the questionnaires and their subscales. These results are presented in Table 2. 

In the Utrecht Work Engagement Scale, the highest score was recorded in the Dedication subscale, whereas the lowest score was in the Absorption subscale. The Vigor subscale had a moderate score. In summary, participants displayed a moderate score in Absorption and a moderate to high score in Vigor and Dedication to their work. Regarding the Maslach Burnout Inventory, participants exhibited a low score in Depersonalization, a moderate score in Emotional Exhaustion, and a high score in Personal Accomplishment. 

Concerning the Utrecht Work Engagement Scale, significant differences were observed as follows: doctors, nurses, and administrative staff scored higher than the other health professionals working in the setting in Vigor (F(3, 146) = 2.771, *p* = 0.044, and partial eta^2^ = 0.054), Dedication (F(3, 146) = 18.826, *p* = 0.001, and partial eta^2^ = 0.279), and Absorption (F(3, 146) = 3.552, *p* = 0.106, and partial eta^2^ = 0.068). Participants with less than five years of experience in the healthcare unit scored higher in Absorption compared to other categories (F(3, 147) = 3344, *p* = 0.021, and partial eta^2^ = 0.064).

Concerning the Maslach Burnout Inventory, significant differences were found as follows. Women had higher scores in Personal Accomplishment compared to men (t = −2.678, df = 149, *p* = 0.008, Cohen’s d = −0.444). Participants with permanent employment contracts had higher scores than those with nonpermanent contracts in Emotional Exhaustion (t = 3.302, df = 149, *p* = 0.001, Cohen’s d = 0.701), Depersonalization (t = 2.941, df = 149, *p* = 0.004, Cohen’s d = 0.625), and Personal Accomplishment (t = 2.320, df = 149, *p* = 0.022, Cohen’s d = 0.493). Doctors, nurses, and administrative staff scored higher in Personal Accomplishment compared to other health professionals working in the setting (F(3, 146) = 11.121, *p* = 0.001, partial eta^2^ = 0.005). Participants with less than five years of experience in the healthcare unit scored lower in Depersonalization compared to other categories (F(3, 146) = 4.297, *p* = 0.006, and partial eta^2^ = 0.081).

Next, we performed correlation analysis, using Pearson’s correlation coefficient, to explore linear correlations among the subscales of the Utrecht Work Engagement Scale and the Maslach Burnout Inventory. These correlations are presented in Table 3. 

The Utrecht Work Engagement Scale revealed the following correlations. Vigor correlated negatively with Emotional Exhaustion and Depersonalization and positively with Personal Accomplishment. Dedication was positively correlated with Personal Accomplishment and negatively with Depersonalization. Absorption displayed a positive correlation with Personal Accomplishment, while it was negatively correlated with Depersonalization and Emotional Exhaustion. The strongest correlation was observed between Personal Accomplishment and Dedication, indicating that higher dedication to work was associated with greater personal accomplishment among healthcare professionals. Conversely, the weakest significant correlation was found between Dedication and Depersonalization. The correlations of Absorption with Emotional Exhaustion and with Depersonalization were also weak. 

In light of the findings mentioned above, three multiple linear regression analyses were conducted using the scores for Emotional Exhaustion, Depersonalization, and Personal Accomplishment as the dependent variable. These analyses revealed no indication of multicollinearity among the variables, as tolerance levels exceeded 0.1 and Variance Inflation Factor (VIF) values were below ten. The examination of Cook and Mahalanobis distance, Centered Leverage Value, and DfBetas and Dffits showed no evidence of outliers or influential points. The results of the multiple linear regression are given in Table 4. 

The results presented above indicate a statistically significant negative association between the score on Vigor and the score on Emotional Exhaustion. Additionally, individuals with permanent contracts displayed higher Emotional Exhaustion. The proportion of variance in the Emotional Exhaustion score accounted for 9.6% (R = 0.347; R-squared = 0.120; Adjusted R-square = 0.096), with the best predictor variable in this model being the score on working status.

Similar outcomes were observed concerning Depersonalization, where the proportion of variance explained by all independent variables was 8.1%. (R = 0.334; R-squared = 0.112; and Adjusted R-square = 0.081), with Vigor being the best predictor variable in this model. 

Concerning Personal Accomplishment, the proportion of variance explained by all independent variables was 24.1%. (R = 0.516; R-squared = 0.267; and Adjusted R-square = 0.241), with Dedication being the best predictor variable in this model. Vigor was also a significant predictor of Personal Accomplishment.

## 4. Discussion

The present study assessed the work engagement and burnout of professionals in a private healthcare unit in Greece. This is a noteworthy contribution as no prior study has explored this topic in this country, making it a valuable addition to the existing literature. Furthermore, this study’s significance is heightened by the fact that it was conducted during the COVID-19 pandemic.

One key finding, which will be elaborated upon below, is the significant negative correlation between work engagement and burnout. In other words, health professionals with high levels of work engagement, particularly those with high Vigor, experienced lower levels of burnout. 

Regarding the Utrecht Work Engagement Scale, participants demonstrated a moderate score in Absorption and a moderate to high score in Vigor and Dedication to their work. These findings align with Kartal’s research, which noted that both private and public healthcare professionals tend to exhibit moderate levels of work engagement [10]. 

The highest score was observed in the Dedication subscale, followed by the Vigor and Absorption subscales. This pattern, which places Dedication as the highest-scoring dimension, slightly differs from the findings in Kartal’s research, in which Dedication also received the highest score but was followed by Absorption and then Vigor [10]. These results emphasize the significance of dedication to work as a prominent dimension of work engagement. 

No significant differences were detected in the participants’ work engagement, based on their demographic characteristics. In contrast to other studies, our findings did not confirm research question 1. 

On the other hand, as is expected, the profession is associated with an engagement in work. Doctors, nurses, and administrative staff had higher work engagement than the other professionals. An additional interesting finding is that those with less than five years of experience in the healthcare unit had higher scores in Absorption. This denotes that individuals with less experience may exhibit a higher level of complete and joyful immersion in their tasks. These findings are different from those found in other studies [7,8]. Thus, research question 2 was partially confirmed. 

In addition, the participants presented a low score in Depersonalization, a medium score in Emotional Exhaustion, and a high score in Personal Accomplishment. Burnout is common among healthcare professionals, as recent studies reveal [17,21,22]. In Greece, moderate levels of burnout have also been found in health workers, although these studies have been conducted in the public health system [24,25,26]. 

Research questions 3 and 4 were partially confirmed. In line with Tsirka, it was found that women had higher scores in Personal Accomplishment compared to men [26]. However, in contrast, Bogiatzaki et al. reported that men had higher scores in Personal Accomplishment [23]. Those with permanent employment contracts had higher scores in all dimensions of burnout (Emotional Exhaustion, Depersonalization, and Personal Accomplishment). Those with less than five years of experience in the healthcare unit had lower scores in Depersonalization. This contrasted with Bogiatzaki et al.’s findings, where those with fewer years of experience had higher burnout levels [23]. It is worth noting that doctors, nurses, and administrative staff had higher scores in Personal Accomplishment compared to other health professionals working in the setting. 

Vigor, Dedication, and Absorption were each negatively correlated with Emotional Exhaustion and with Depersonalization and were each positively correlated with Personal Accomplishment. Similar results have been reported by Panari et al. [33]. This aligns with Schaufeli and Bakker’s argument that interventions strengthening work engagement can reduce burnout [31]. Other researchers have also emphasized the crucial role of work engagement in preventing burnout, especially in the context of the COVID-19 pandemic [11,26]. Based on these results, research question 5 was confirmed. Work engagement is positively associated with resilience and a sense of significance derived from one’s work, making it a significant protective factor that aids healthcare professionals in coping with burnout.

The strongest correlation was observed between Dedication and Personal Accomplishment. This outcome aligns with expectations and underscores the crucial role of Dedication to work in preventing burnout. Conversely, the weakest significant correlation was identified between Dedication and Depersonalization, suggesting that dedication to work has a comparatively lower impact on the depersonalization dimension of burnout. Notably, no significant correlation was found between Dedication and Emotional Exhaustion. Additionally, multiple linear regression analyses revealed that Vigor was a significant predictor of Emotional Exhaustion, Depersonalization, and Personal Accomplishment. The reason for this may have to do with the fact that this study was conducted during the COVID-19 pandemic. High levels of energy and mental resilience during work in this context, and during health crises in general, may be significant protective factors against burnout. This is an interesting phenomenon that should be examined in future studies. Dedication was also a significant predictor of Personal Accomplishment. These findings underscore the intricate relationships between work engagement and the various dimensions of burnout syndrome and highlight the significant role of Vigor in burnout prevention. 

It is essential to note that this study was conducted during the COVID-19 pandemic. Although the healthcare unit was not a frontline COVID-19 hospital, healthcare professionals were exposed to heightened risks, including the increased risk of infection for themselves and their families, as well as the burden of making challenging decisions. This context provides valuable insights into the strong negative correlation between work engagement and burnout, emphasizing the significance of work engagement in maintaining the well-being of healthcare professionals. 

Strengths of this research include its originality for studying Greek health professionals of private health units and the fact that it involved a heterogeneous sample, encompassing various professions. A primary limitation is the cross-sectional design of this study, which does not allow for the establishment of causal relationships. Additionally, this study’s focus on Greek society and its restriction to a single private healthcare unit may limit the generalizability of the findings. However, it is worth noting that the limited number of great private hospitals in Greece makes this sample reasonably representative. Moreover, it is important to consider that this study was conducted during the COVID-19 pandemic, which could have influenced the reported burnout syndrome and work-engagement dimensions. 

Despite the discussed limitations, these results hold important practical implications and convey a clear message to hospital managers and policymakers. They highlight the need to take this issue seriously and provide recommendations for supporting and promoting work engagement to prevent and address burnout. Special attention should be given to more vulnerable groups of professionals, such as women and those with permanent employment contracts. 

It is recommended that future research further investigates the demographic and work-related characteristics influencing work engagement and burnout among professionals in private healthcare units, ideally with a larger sample size. Comparing these variables between the private and public health sectors could provide insights into significant differences. Similar studies could be conducted in different countries or regions to explore these issues, particularly in the post-COVID-19 pandemic era. Additionally, the evaluation of work engagement and burnout could benefit from longitudinal study designs or from focusing on specific professional groups, such as doctors and nurses, working in private healthcare units. 

## 5. Conclusions

The findings indicate that healthcare professionals in the private health sector in Greece experience moderate levels of burnout and exhibit moderate work engagement. Notably, there was a negative association between work engagement and burnout. Additionally, various significant differences in professionals’ burnout and work engagement were identified, based on their sociodemographic and work-related characteristics. It is crucial to promote work engagement, especially the dimension of Vigor as it is positively associated with resilience, to prevent and address burnout among healthcare professionals.

## Figures and Tables

**Table 1 ijerph-21-00130-t001:** Demographic characteristics of the sample and work-related information.

		Frequency	%
Sex	Man	61	40.4%
	Woman	90	59.6%
Age	18–25 years old	22	14.6%
	26–35 years old	40	26.5%
	36–45 years old	44	29.1%
	>46 years old	45	29.8%
Marital status	Married without children	14	9.3%
	Married with children	57	37.7%
	Single	67	44.4%
	Separated/divorced/widower	13	8.6%
Number of children	0	86	57.0%
	1	13	8.6%
	2 or more	52	34.4%
Level of education	Post-secondary education graduate	50	33.1%
	University/Technological Educational Institute graduate	72	47.7%
	M.Sc./Ph.D. Holder	29	19.2%
Income/month	Up to 800 €	61	40.4%
	801–1300 €	71	47.0%
	>1301 €	19	12.6%
Working status	Permanent contract	124	82.1%
	Nonpermanent contract	27	17.9%
Years of work in the healthcare unit	<5 years	47	31.1%
	6–10 years	28	18.5%
	11–15 years	38	25.2%
	>16 years	38	25.2%
Profession	Nurse	54	36.0%
	Doctor	18	12.0%
	Administrative Staff	53	35.3%
	Other health professionals working in the setting	25	16.7%

**Table 2 ijerph-21-00130-t002:** Descriptive statistics for the questionnaires and their subscales.

	Mean	SD	Min	Max
Utrecht Work Engagement Scale				
Vigor	3.278	0.821	1	5
Dedication	3.443	0.919	1	5
Absorption	3.064	0.845	1	5
Maslach Burnout Inventory				
Εmotional Exhaustion	25.113	9.365	2	50
Depersonalization	7.934	4.827	0	26
Personal Accomplishment	32.358	8.139	5	46

**Table 3 ijerph-21-00130-t003:** Correlations between the Utrecht Work Engagement Scale and the Maslach Burnout Inventory.

	Utrecht Work Engagement Scale			Maslach Burnout Inventory		
	Vigor	Dedication	Absorption	Εmotional Exhaustion	Depersonalization	Personal Accomplishment
Utrecht Work Engagement Scale						
Vigor	1					
Dedication	0.704 **	1				
Absorption	0.817 **	0.782 **	1			
Maslach Burnout Inventory						
Εmotional Exhaustion	−0.249 **	−0.080	−0.190 *	1		
Depersonalization	−0.287 **	−0.177 *	−0.201 *	0.495 **	1	
Personal accomplishment	0.356 **	0.435 **	0.269 **	−0.059	−0.193 *	1

Note: *—Correlation is significant at the 0.05 level (2-tailed). **—Correlation is significant at the 0.01 level (2-tailed).

**Table 4 ijerph-21-00130-t004:** Multiple linear regression with the score in Emotional Exhaustion. Depersonalization, and Personal Accomplishment as the dependent variable.

Dependent Variable = Emotional Exhaustion							
Predictor	Unstandardized Coefficients		Standardized Coefficients	T	Sig.	95.0% Confidence Interval for B	
	B	Std. Error	Beta			Lower Bound	Upper Bound
(Constant)	38.211	3.735		10.231	<0.001	30.829	45.592
Working status Permanent contract * versus nonpermanent contract	−5.180	1.947	−0.213	−2.660	0.009	−9.027	−1.332
Years of work in the healthcare unit <5 years or 6–10 years * versus 11–15 years or >16 years	0.182	0.362	0.050	0.487	0.688	−0.580	0.965
Vigor	−3.870	1.592	−0.339	−2.430	0.016	−7.017	−0.723
Dedication	0.884	1.305	0.087	0.677	0.499	−1.696	3.464
Absorption	0.869	1.729	0.078	0.503	0.616	−2.548	4.286
**Dependent Variable = Depersonalization**							
(Constant)	14.210	2.467		5.761	<0.001	9.335	19.086
Working status Permanent contract * versus nonpermanent contract	−2.286	1.096	−0.182	−2.085	0.039	−4.453	−0.119
Years of work in the healthcare unit <5 years or 6–10 years * versus 11–15 years or >16 years	0.164	0.351	0.040	0.466	0.642	−0.531	0.859
Vigor	−2.014	0.829	−0.343	−2.431	0.016	−3.652	−0.377
Dedication	−0.309	0.681	−0.059	−0.454	0.651	−1.654	1.036
Absorption	1.205	0.899	0.211	1.340	0.182	−0.573	2.982
**Dependent Variable = Personal Accomplishment**							
(Constant)	0.344	0.330		1.045	0.298	−0.307	0.996
Working status Permanent contract * versus nonpermanent contract	−0.154	0.147	−0.083	−1.051	0.295	−0.444	0.136
Years of work in the healthcare unit <5 years or 6–10 years * versus 11–15 years or >16 years	0.084	0.047	0.138	1.779	0.077	−0.009	0.176
Vigor	0.298	0.111	0.345	2.694	0.008	0.079	0.517
Dedication	0.392	0.091	0.508	4.315	<0.001	0.213	0.572
Absorption	−0.325	0.120	−0.387	−2.703	0.008	−0.562	−0.087

Note: * Reference category, Number of observations = 151.

## Data Availability

The data presented in the study are available on request from the corresponding author. Data available on request due to restrictions.

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
