# Peer review of "Work Engagement and Burnout in a Private Healthcare Unit in Greece"

_ijerph, 2024, doi:10.3390/ijerph21020130_

Round 1

Reviewer 1 Report (Previous Reviewer 4)

Comments and Suggestions for Authors

This article is a manuscript that has been resubmitted by the authors after multiple revisions. Indeed, the readability and correctness of this manuscript have been improved after numerous revisions. However, this manuscript still needs further revision. Therefore, I offer the following review comments.

1.      In line 154, 161 and 175, authors should add the subsection number (e.g., 2.3.1, 2.3.2 and 2.3.3).

2.      All Tables must be labeled and described in the text. Except for Table 1, the other three Tables lack specific markup of the content. The authors even lack explanation for the text of Table 2. Please correct them.

3.      On the multiple regression analysis, there are the following review comments.

(1)   Why did the authors not analyze regression analysis with personal accomplishment as the dependent variable?

(2)   All regression analysis results point to only vigor being significant. This is an interesting phenomenon. The reason may have to do with the fact that this study was conducted during COVID-19. I suggest that the authors discuss this phenomenon in depth.

4.      Please arrange all Tables on the same page.

5.      In line 126, 128 and 276, please add a period at the end of the sentence.

Comments on the Quality of English Language

I think the authors should recheck the English usage throughout the manuscript because the authors often use “Furthermore”, “it is noted that …”, “it is noteworthy”. Although this usage is not wrong, it may be boring to readers. I suggest that the authors make appropriate corrections.

Author Response

Dear reviewer, 

Thank you very much for your valuable feedback. You can see our responses in the attached file. 

Best regards!

Reviewer 2 Report (New Reviewer)

Comments and Suggestions for Authors

Thanks to the authors for this new version on the manuscript and their trust in the journal IJREPH.

Still there are some points to be improved in this new version of the manuscript. 

General comments: The subject matter of this study is undoubtedly relevant to the health and personal growth for the health professionals

ABSTRACT: 

Abstract doesn´t reflects the best of the paper. Please re-write and have a look at the typo-corrections that needs.

One of the major problems in this study is the variables included: "vigor and dedication" and how they have been measured. Both questions are in some way new, but not clear enough in the abstract, because it is first mentioned here. What means “vigor and dedication”?

Later at the introduction it is provide these 2 questions more clear than in the abstract. So abstract should be re-written. Please describe properly that Vigor, Dedication and  Absorption are subdimensions of the Utrecht Work Engagement Scale.

Words as: “quantitative primary”- not necessary primary, it can be describe as a “quantitative cross-sectional with health professionals working in contact with patients; doctors, nurses, administrative and others involved in the context

When explaining the sample and participants avoid vague description as: "other professionals" I will suggest to substitute by: "other health professionals involved in the context".

I do agree with reviewers that the heterogeneity of participants affects the external validity or generalization of results. To add, that readers we do not have any idea about which is the proportion of population that compounds professionals in this sector. Not even from doctors, nurses or other health professionals. You may want to frame this in relation to each professional category or at least some approx. data that provides some context.

About the type of contract it will be beneficial to describe as: "permanent and nonpermanent contract". Definitive is not the most appropiate term.

Please, in the abstract, and, because is the initial description of: "Concerning work engagement, the participants demonstrated a moderate score in absorption and medium to high scores in vigor and dedication". Definition on what means "absortion" - vigor or dedication is needed. Not clear. In the main text authors explain both terms better so, I will suggest re-write abstract differently and, more focus in abstract only with the most important content.

At the end of the introduction please delete the phrase: “The study is structured as follows: first, we present the methodology. Then, we present the results and discuss them. Finally, we discuss the clinical implications of the study, its strengths and limitations, and suggest directions for future research. It’s not necessary and do not add anything to the text.

Regarding the hypothesis and, because is a descriptive study design it is not necessary but I could agree to be stated in text as conceptual hypothesis or research questions.

METHODS: 

Sampling: please explain, if possible the total population – see comment above

Measures: the data collection was done with a questionnaire. Please delete the word composite. Not necessary

It will be useful to add citation/ references about the instruments Utrecht Work Engagement Scale, and the Maslach Burnout Inventory, scales that have been translated and validated in the Greek population, also add year of validation. The citation number 37 is not for the Utrecht Work Engagement Scale, please correct (it is the number 3- but for the English version of the scale)

RESULTS. 

Please do not repeat some of the content of tables. In results section only main data has to be described and supported by tables.

DISCUSSION:

THE SECOND PRAGRAPH in the section of discussion it's quite explanatory about the Health system in Greece and I should recommend to wirte in the introduction and not in discussion.

In Greece, the number of practitioners in private medical institutions is relatively small, 299 with the majority of hospitals falling under the public sector. Private hospitals are primarily 300 concentrated in the capital city, Athens, and encompass both general and specialized facilities, 301 such as maternity or psychiatric care clinics. Additionally, there are smaller private healthcare 302 units, known as "clinics," offering specialized therapeutic care, such as general surgery or 303 childbirth services. Similar to other countries, Greece exhibits significant differences in the 304 organization, administration, and operation of public and private hospitals. Public hospitals 305 provide primary, secondary, and tertiary healthcare, employing civil servants, while private 306 hospitals mainly deliver secondary healthcare, and their professionals hold contracts for definite periods or serve as visiting doctors.

It should be done some reference about the data

Author Response

Dear reviewer, 

Thank you very much for your valuable feedback. You can see our responses in the attached file. 

Best regards!

Round 2

Reviewer 1 Report (Previous Reviewer 4)

Comments and Suggestions for Authors

After many revisions, the authors have adjusted all the review comments and improved the readability and accuracy of this manuscript.

Author Response

Dear reviewer, 

Thank you very much for your comments and your Kind response. 

Best regards!

This manuscript is a resubmission of an earlier submission. The following is a list of the peer review reports and author responses from that submission.

Round 1

Reviewer 1 Report

Comments and Suggestions for Authors

Dear authors, thank you for the opportunity to get acquainted with your interesting research.

The topic of the study is relevant and allows filling the knowledge gap regarding the impact of involvement in work on the severity of emotional burnout of employees of a private medical institution. In the introduction, the authors convincingly showed that although the issues of work involvement and burnout are not new, a detailed study of their relationship has not been sufficiently studied. A clear statement of the purpose and hypotheses of the study is very valuable, and the alignment of the presentation of the article itself is structured according to these key provisions.   The study sample itself is a strong point in the study, because it was attended by medical staff of various specialties, work experience, education, age. The authors clearly present all these characteristics in the table.   In their study, the authors, on the one hand, showed the severity of emotional burnout and involvement in work among medical workers of various specialties, professional experience, age, and on the other hand, revealed the features of the relationship between the various components of emotional burnout and involvement in work, showed the specifics of their relationship.   As a possible addition, we can suggest the authors to apply one of the multivariate methods of statistical data processing in order to enhance the effect that we saw from the results of the correlation analysis. The discussion by the authors is conducted very competently. the results of each hypothesis of the study are discussed in detail, correlations with the results of other authors are given, the strengths and limitations of the study are highlighted, and prospects for further research are given. The authors may slightly expand the limitations of the study regarding the possibility of comparison with a sample from a public medical institution, as well as the use of other methods that showed slightly different results in studies of other authors (which the authors write about in the analysis of other studies).   Conclusions based on the results of the work are relevant and concise. All references refer to the study, and I would also like to note a fairly high percentage in the list of references of new sources (for the last 3-5 years), which indicates a good study of the issue by the authors. The tables presented by the author are very clear and illustrative. The strength is that the authors were able to succinctly and fully present all the results of their research.

Best regards, reviewer

Author Response

Dear reviewer, 

see the attached file. 

Thank you very much!

Reviewer 2 Report

Comments and Suggestions for Authors

The theme of this paper is interesting, and certainly useful for society, but:

ABSTRACT

The abstract is what attracts (or does not) the attention and interest to the article. So, it should be carefully written.

Authors should improve the writing of the purpose of the article so as not to become so subjective.

The authors should improve the writing of the main results obtained and mention that this is a case study in a private hospital in Greece.

1.       INTRODUCTION

An introduction should be informative and well-worded. Therefore, it is missing:

1-    present the theoretical problem/formulation and the objective of the study.

2-    give clues to the discussion of the results.

3-    present the structure of the article.

2.       METHODS

This chapter shall provide the necessary and sufficient information to assess how the study was conducted in order to allow its reproduction by other.

There is no presentation of the questionnaire. It would be interesting to put the questionnaire in Appendix, so that other researchers replicate the study.

It is not known how the questionnaire it was answered. Respondents were online or in person.

All this relevant information is missing.

Between point 2. Methods and point 6. Results there are too many points.

I suggest they be grouped together: some in point 2. Methods and others in point 6. Results

I also suggest that the point of Results be Results and Discussion

6.       RESULTS

This point should be articulated with the previous one, as the results presented should be supported by the methods.

The text should be improved. It's all very confusing.

The results should show the evidence of the study and should be presented according to a logical and informative and perceptible sequence.

It is not understood which methods are used to obtain the results, i.e., explain the methodology that was use.

It is not perceived how and when the study was implemented.

More than describing the tables and data, it is necessary to reflect and discuss the results obtained. This point should be improved.

7. DISCUSSION

The theoretical implications and possible practical applications should be discussed.

The authors should mention that this is a study based on the reality of Greece.

The originality and relevance of the research presented should be strengthened.

This point should be improved and there should be a link to the theory presented in the Introduction. The way in which the results relate to each other and with those of other studies should be evidenced.

Limitations and Future research should be at the end of the conclusions!

Authors should avoid repeating the same ideas at different points in the text.

This point should be improved.

5. CONCLUSION

The results should be presented in a comprehensive manner, highlighting the most relevant ones and provide a summary of the text.

The originality and relevance of the results presented should be strengthened.

The limitations of the study itself should be presented and discussed here and not at a previous point.

Another limitation of this research is that the study focuses only on Greece society.

The future investigation should be put here at the end and needs to be more developed. One suggestion for future research: conduct the same study in other countries/regions of the world. Or in another hospital, but public. Or it would be interesting to conduct the same study regarding what happened in the post-pandemic situation.

Author Response

(The authors gave the same response as above.)

Reviewer 3 Report

Comments and Suggestions for Authors

 It was my pleasure to review this manuscript, which deals with work engagement and burnout in a private health service unit in Greece. The manuscript aims to evaluate the work engagement and burnout of professionals in a private healthcare unit and to identify the demographic and work-related characteristics that affect them. A cross-sectional study with a convenience sampling method using a structured questionnaire was conducted in a private healthcare unit in the southern suburbs of Athens, Greece. The study analyzed a total of 151 employees from different professional specialties, including doctors, nurses, administrative staff, medical laboratory technicians, and computer science specialists. In summary, I found the topic to be quite interesting and worth investigating, especially during the pandemic period. However, with the sole objective of improving the quality of the manuscript, I will allow myself to make a few comments.

1.      Introduction:

The introduction section of the manuscript would benefit from a more precise and structured approach. Currently, it consists of 15 paragraphs, which leads to a lack of centralized focus. To improve the quality of the introduction, it is recommended to restructure the content and specifically addresses prior findings and evidence related to work engagement, burnout status, and work-related characteristics among healthcare professionals.

2.      Method

The method section is well-structured and provides vital information.

3.      Results (page 7, line 220)

Please correct the formatting of this sentence because it should be a new paragraph.

4.      Results (page 7, table 4)

There is one mistake in the note. Instead of "*Correlation is significant at the 0.01 level (2-tailed)," the authors can rephrase it as "*Correlation is significant at the 0.05 level (2-tailed)."

5.      Discussion

The authors mentioned that “The current study evaluated the work engagement and burnout of professionals in a private healthcare unit in Greece. No other study has examined this topic in this country, thus highlighting its novelty.” However, as this study investigates work engagement and burnout among healthcare professionals during the pandemic period, the manuscript could incorporate additional informative messages regarding the experiences within this specific healthcare unit. For example, it could explore whether healthcare workers are more susceptible to physical and mental exhaustion due to the torment of difficult decisions, the emotional toll of losing patients and colleagues, and the increased risk of infection for themselves and their families. This would enable readers to gain a deeper understanding of how work engagement during the pandemic period impacts healthcare workers' perception of burnout and related aspects. Furthermore, by incorporating such information into the revised manuscript, it would not only provide valuable insights into the experiences within this specific healthcare unit but also highlight the novelty of this study in exploring the unique challenges faced by healthcare professionals during the pandemic period, shedding light on the impact of work engagement on their perception of burnout and related aspects.

Author Response

(The authors gave the same response as above.)

Reviewer 4 Report

Comments and Suggestions for Authors

Work engagement and burnout are indeed serious issues for healthcare practitioners. This is also a medical management issue that has been concerned for a long time. The authors explore this issue with practical value. However, there are still standards for reviewing international academic articles. Therefore, I propose the following review comments for this manuscript.

1.      In my overall evaluation of this manuscript, I find it difficult to find specific contributions from the authors that go beyond past research. There have been many studies on this topic. The authors did not draw from gaps in past research in their motivation for undertaking this study. This makes the research design merely repeat the conclusions of past studies. In addition, the authors' research framework is too simple and does not break through past thinking. Although the statistical methods of t-test, ANOVA, and correlation analysis can still provide academic information, the phenomena that can be explained by the above statistical methods are limited. This should be considered at the outset of research design. Again, since this is a long-standing healthcare management issue, the authors should draw valid hypotheses from past studies. Comparisons with other studies internationally are also necessary in an academic article. Finally, the entire manuscript is a description of the phenomena and does not explore the reasons for these phenomena. This needs to be supplemented. All of the above are lacking in this manuscript. I suggest that the authors think carefully about the contribution of this manuscript again, and re-design, arrange and analyze the survey data before resubmitting. However, although I offer an overall evaluation of this manuscript, I still make my suggestions for the content of this manuscript one by one.

2.      In the selection of research subjects, although the authors surveyed all types of hospital employees, the work stress of these employees varies greatly. In other words, the data collected by the authors was very heterogeneous. It is not appropriate to combine these data for analysis.

3.      Although 151 samples are acceptable in the analysis for comparison of means. In practice, however, too small a sample size will lead to less confidence. After all, the number of practitioners in private medical institutions is very large in Greece. Collecting only 151 samples is not enough in this case. Furthermore, the authors emphasize in the title that the data come from private medical institutions. In other words, the authors argue that private and public healthcare institutions are different. How do they behave differently? The authors should clearly state their differences from the past literature.

4.      In line 14, the authors state that SPSS was used as the analysis tool. In this paragraph, the choice of statistical method is more important than the choice of analytical tools. I suggest that the authors should add a description of the statistical methods (perhaps the authors can also consider removing the description of the analysis tools since the statistical methods of this manuscript are simple).

5.      In the “Abstract” section, in addition to explaining the research results, the authors should also propose the reasons for this phenomenon. This is the most important content of an academic article. However, in the "Discussion" section of the manuscript, the reasons for the findings were not discussed. Please add them.

6.      In line 47-56, the description of this paragraph is inappropriate. I offer the following reasons.

(1)   If the differences in research results are due to different measuring instruments, the data should not be compared. To compare data from different conditions, the research base must be the same.

(2)   The level of scoring and investment is an abstract concept. Under what circumstances is it defined? This requires objective criteria. This descriptive style can be found throughout the manuscript. The authors should make appropriate corrections. In addition, different medical positions have different work pressures, so the resulting work burnout is also different. This needs to be distinguished. Authors should also consider differences caused by studies in different countries (possibly due to institutional differences). I suggest that authors can refer to the analysis literature of meta-analysis.

7.      In line 63-64, the authors make the following description:

Furthermore, it is noted that the importance of engaged healthcare staff was accentuated during the Covid-19 pandemic [13].

The literature citations for this description are inappropriate because the 13th literature was published in 2016. At this time COVID-19 has not yet occurred. Also, "Covid-19" should be corrected to "COVID-19".

8.      In line 67-68, the authors make the following description:

On the other hand, burnout has been recognized as a syndrome and consists of three dimensions (emotional exhaustion, depersonalization, personal accomplishment).

Please indicate the source of the literature citation.

9.      In lines 82-89, I suggest that the authors combine these two paragraphs.

10.  In line 94-97, please explain the reasons for these phenomena.

11.  The three assumptions made by the authors require a rigorous inference process. In other words, the authors should have clear literature citations to infer hypotheses, not just past phenomena that prove the same should be the case in Greece. It is important to extrapolate hypotheses from past research. Demographic differential analysis should infer hypotheses separately, and hypotheses should be made separately by category. In addition, it is inappropriate to assume the significance of the correlation coefficient, because the significance that the correlation coefficient can explain is low.

12.  In Table 1, are the categories with a small number of individuals merged during the analysis process?

13.  In lines 129, 178 and 189, the “3. Measures”, “4. Data analysis” and “5. Ethics” sections should all be part of the “2. Methods” section.

14.  In subsection 3.2, please combine them into one paragraph to describe. In subsection 3.3, please also handle it in the same way.

15.  There are many types of Post hoc test methods. Why did the authors choose “Bonferroni”?

16.  Please add a description of the IRB approved unit.

17.  In Table 2, I make the following suggestions.

(1)   The numbers for Mean and SD should be expressed with three decimal places.

(2)   The authors measure the Maslach Burnout Inventory using a 7-point Likert scale. Therefore, a mean and SD exceeding 7 is unreasonable. Please check again. In line 207-209, the mean and SD figures are also corrected.

(3)   The maximum value for “Absorption” is 4.83. Why is this number not an integer?

(4)   What is the meaning of "Range"? I suggest removing them if it doesn't make sense.

18.  In line 212-233, please describe the research results in full text. Columnar expressions are not an appropriate writing strategy. Also, in line 220-221, this is a separate paragraph.

19.  In line 251-252, the authors make the following description:

No other study has examined this topic in this country, thus highlighting its novelty.

The value of academic articles lies in whether they have academic contributions. If there is no research result that goes beyond the past, even if it has not been studied, it is just a repetition of a known phenomenon in the past. Therefore, I suggest that the authors revise the description of this sentence

20.  In the “Discussion” section, the authors did not discuss the results in depth. That the past phenomenon is the same as this study is not the core of the discussion. The reason for this phenomenon is the important inspiration. This is what this manuscript lacks most. In addition, correlation analysis does not provide much information. Please correct and supplement the authors.

21.  In the format, I make the following suggestions.

(1)   “e.g.,” is the correct expression (lines 30, 83, 126, 144, 145, 146, 160, 162, 165, 317).

(2)   “etc.” is the correct expression (lines 126, 174).

(3)   All statistical results should be expressed with three decimal places.

(4)   In line 42, “[6, 7]” → “[6-7]”.

(5)   In line 68, “[15, 16, 17]” → “[15-17]”.

(6)   In line 79, “[21, 22, 23, 24, 25, 26]” → “[21-26]”.

(7)   In line 89, “[29, 30]” → “[29-30]”.

(8)   In line 95, “[31, 32, 33]” → “[31-33]”.

(9)   In Table 1, the line under “Sex” is redundant.

(10)   In line 145, there is an extra space after “eg.”.

(11)   In Table 4, please adjust the typesetting of “Note”. Also, please delete the horizontal line at the bottom.

(12)   In line 271, “[15, 16, 17]” → “[15-17]”.

(13)   In line 274, “[21, 25, 26]” → “[21, 25-26]”.

(14)   In line 274, “[26, 27, 28]” → “[26-28]”

(15)   In line 284, there is an extra space after "[27]".

(16)   In line 291, “[14, 29, 30]” → “[14, 29-30]”.

(17)   In line 314, there is an extra space after “unit.”.

(18)   In line 323, there is an extra space after “characteristics.”

Comments on the Quality of English Language

This manuscript has many usage errors in the use of English, and there are also grammatical idioms that do not conform to English. I suggest that authors ask native English-speaking professionals to re-examine the entire manuscript.

Author Response

(The authors gave the same response as above.)

Reviewer 5 Report

Comments and Suggestions for Authors

Introduction

1. The first sentence is a complete truism and should be deleted. Also repeated in lines 106-107 - should be deleted there as well. 

2. Negative correlation between burnout and work engagement is mentioned couple of times in Introduction. Introduction section could be better organized and this correlation should be mentioned only once with all references needed to prove this statement. 

3. Lines 67-68 - "Higher dedication is reported by women, doctors and nurses, (...)." Higher dedication reported by doctors and nurses than by...(which profession? midwives?). 

4. Lines 74-76 - organizational level of burnout and its meaning for individual perceptions should be better explained. 

5. Lines 88-89 - Which populations were considered in these COVID-19 oriented studies? 

6. Lines 92-93 - Maybe it would be better to mention particular professions here (doctors, nurses) than university vs. secondary school graduates? 

7. Lines 96-97 - Which specialties are mentioned here?

8. Introduction section should be shortened and better organized. 

Methods

1. Since the project was carried out during the COVID-19 pandemic,  some data about burden of the outbreak for the healthcare sector in Greece should be presented (in Introduction, probably) - number of cases overall, number of deaths...

2. More details about the studied unit should be given - is that general practice? Specialty care? Private out-patient clinic or hospital? 

3. Authors mentioned, that the scales used were "culturally adapted in the Greek population" - what does it mean? Which adaptations were made?

4. Using word "specialty" for professions like doctors, nurses, etc. is not a good choice. Specialty refers more to medical domains like psychiatry, gynecology...I would rather use word "profession".  

5. The information about acceptance received from the Local Ethics Committee should be given in the "5. Ethics" section. 

Results

1. I can not understand where did the Minimum and Maximum scores of UWES come from (Table 2.)? Is it not a sum of particular statements in each subscale? If so, the minimum in each subscale should be 5, right? Or is this arithmetic mean? Since in MBI it is sum, this issue should be clarified. 

2. Lines 202-204 is just a repetition of what is showed in the Table 2. Similarily, lines 206-209. 

3. I am not sure if Table 3. is needed, since all significant findings are enumerated below - lines 212-233. I would submit Table 3 as Supplementary Material. Similarly, I would not repeat findings presented in Table 4 (lines 237-243). 

4. Table 4 - I think that * is 0.05 not 0.01, right? 

Discussion

1. Authors should better explain why they did choose private sector employees for this study? What are differences in organizational factors and institutional culture between private and public sector in Greece? 

2. I am not sure if the study gives recommendations for policy makers (lines 309-310).  What kind of recommendations? Study of such a low generelizability can not be background for any recommendations and policies. 

Conclusions

1. I am not sure if authors proved that work engagement is protective factor preventing burnout. 

General remarks

The most important concern about the study is its extremely low generalizability. Study population of only one private institution gives actually no chance to draw any conclusions. Some organizational factors in this particular institution may influence results very much. 

Author Response

(The authors gave the same response as above.)

Round 2

Reviewer 2 Report

Comments and Suggestions for Authors Congratulations

Reviewer 4 Report

Comments and Suggestions for Authors

After the authors' revisions, the readability of this manuscript has indeed been improved. But there are still many flaws that need to be corrected. Therefore, I still submit my review comments on this revised manuscript.

1.      In point 2 of the last review comments, I questioned the heterogeneity of the authors' data collection. In their responses, the authors pointed out that data heterogeneity is not only good but also an advantage. I don't agree with this answer. If the authors considered this a research strength, different employee types should be compared separately. However, it is a pity that the number of samples in each occupational category is very small. If compared separately, it will cause doubts about low credibility. I think this is an irreparable flaw in the research.

2.      In the “Abstract” section, it is the description of the research method that is important rather than which research tool was used. If the authors still think it is important to write research instruments in the “Abstract” section, I have no objection. However, describing the research methods is an indispensable part of this section. This opinion has been raised in the previous review comments. It is a pity that the authors did not respond appropriately.

3.      In point 5 in the previous review comments, my suggestions are as follows:

“In the “Abstract” section, in addition to explaining the research results, the authors should also propose the reasons for this phenomenon. This is the most important content of an academic article. However, in the "Discussion" section of the manuscript, the reasons for the findings were not discussed. Please add them.”

The authors did not respond to this review comments. Authors are requested to provide appropriate feedback.

4.      In line 38-48, the description in this paragraph is common sense. The performance of work burnout is different among doctors and nurses in different medical fields. This conclusion has been confirmed by many studies. However, the authors use unified concepts to explain professional burnout among medical personnel. This is not refined enough.

5.      In point 7 in the previous review comments, my suggestions are as follows:

““Furthermore, it is noted that the importance of engaged healthcare staff was accentuated during the Covid-19 pandemic [13].

The literature citations for this description are inappropriate because the 13th literature was published in 2016. At this time COVID-19 has not yet occurred. Also, “Covid-19” should be corrected to “COVID-19”.”

In this response, the authors referred to deleting the literature citations. This is inappropriate handling. I suggest that authors still provide appropriate literature citations.

6.      In point 11 in the previous review comments, my suggestions are as follows:

“The three assumptions made by the authors require a rigorous inference process. In other words, the authors should have clear literature citations to infer hypotheses, not just past phenomena that prove the same should be the case in Greece. It is important to extrapolate hypotheses from past research. Demographic differential analysis should infer hypotheses separately, and hypotheses should be made separately by category. In addition, it is inappropriate to assume the significance of the correlation coefficient, because the significance that the correlation coefficient can explain is low.”

The authors did not respond to this review comments. Authors are requested to provide appropriate feedback.

7.      When the authors did t-test and ANOVA analysis, there were still many categories with sample size less than 10 that were not properly processed. Please correct them.

8.      The authors believe that it is appropriate to use “Bonferroni” for post hoc testing because this method is highly flexible, simple to calculate, and does not consider the number of samples in different categories. What does it mean to be highly flexible? Is "computational simplicity" an issue that should be considered? There are many post hoc testing methods that do not consider the number of samples in different categories. What are the differences between these methods? Please explain.

9.      In Table 2, it is customary for general statistical methods to use averages for calculation. If you want to calculate by totaling, please specify.

10.  In point 18 in the previous review comments, my suggestions are as follows:

“In line 212-233, please describe the research results in full text. Columnar expressions are not an appropriate writing strategy. Also, in line 220-221, this is a separate paragraph.”

The authors did not respond to this review comments. Authors are requested to provide appropriate feedback.

11.  Please place the content at the top of all table fields to avoid misunderstanding by readers.

12.  Authors are asked to arrange the contents of Table 1 on the same page.

13.  I don’t understand what the authors propose as the purpose of Table 4? I think this is inappropriate for the following reasons. First, this table is not explained in this article. Secondly, small samples are used under the analysis tool of SPSS, and the results are unstable. In other words, the analysis results are not trustworthy. Finally the control variables are used blindly and are not explained in the “Methods” section.

14.  Please include a description of the statistical methods used in the description of the “Data analysis” subsection.

15.  Please add section and subsection numbers according to IJERPH journal format regulations.

16.  The "p" value should be expressed in lowercase and italics.

17.  The expression “T-test” should be revised to “t-test”.

18.  In a good academic paper, the “Results” section and the “Discussion” section should be separated.

19.  The spacing in P. 8-10 is wrong.

Comments on the Quality of English Language

I do not think this manuscript is perfect in its use of English because there are still many questionable expressions. I still suggest that authors ask a native English-speaking professional to review the entire manuscript.

Author Response

(The authors gave the same response as above.)
